# Gate Decorator: Global Filter Pruning Method for Accelerating Deep Convolutional Neural Networks

**Zhonghui You** [1]
Peking University
zhonghui@pku.edu.cn

**Kun Yan** [1]
Peking University
kyan2018@pku.edu.cn

**Jinmian Ye**
Momenta
jinmian.y@gmail.com

**Meng Ma** [2, *]
Peking University
mameng@pku.edu.cn

**Ping Wang** [1, 2, 3, *]
Peking University
pwang@pku.edu.cn

[1] School of Software and Microelectronics, Peking University
[2] National Engineering Research Center for Software Engineering, Peking University
[3] Key Laboratory of High Confidence Software Technologies (PKU), Ministry of Education

## Abstract

Filter pruning is one of the most effective ways to accelerate and compress convolutional neural networks (CNNs). In this work, we propose a global filter pruning algorithm called *Gate Decorator*, which transforms a vanilla CNN module by multiplying its output by the channel-wise scaling factors (*i.e.* gate). When the scaling factor is set to zero, it is equivalent to removing the corresponding filter. We use Taylor expansion to estimate the change in the loss function caused by setting the scaling factor to zero and use the estimation for the global filter importance ranking. Then we prune the network by removing those unimportant filters. After pruning, we merge all the scaling factors into its original module, so no special operations or structures are introduced. Moreover, we propose an iterative pruning framework called *Tick-Tock* to improve pruning accuracy. The extensive experiments demonstrate the effectiveness of our approaches. For example, we achieve the state-of-the-art pruning ratio on ResNet-56 by reducing 70% FLOPs without noticeable loss in accuracy. For ResNet-50 on ImageNet, our pruned model with 40% FLOPs reduction outperforms the baseline model by 0.31% in top-1 accuracy. Various datasets are used, including CIFAR-10, CIFAR-100, CUB-200, ImageNet ILSVRC-12 and PASCAL VOC 2011.

## 1 Introduction

In recent years, we have witnessed the remarkable achievements of CNNs in many computer vision tasks [40, 48, 37, 51, 24]. With the support of powerful modern GPUs, CNN models can be designed to be larger and more complex for better performance. However, the large amount of computation and storage consumption prevents the deployment of state-of-the-art models to the resource-constrained devices such as mobile phones or the Internet of Things (IoT) devices. The constraints mainly come from three aspects [28]: 1) Model size. 2) Run-time memory. 3) Number of computing operations. Take the widely used VGG-16 [39] model as an example. The model has up to 138 million parameters and consumes more than 500MB storage space. To infer an image with resolution of $224 \times 224$, the model requires more than 16 billion floating point operations (FLOPs) and 93MB extra run-time

---

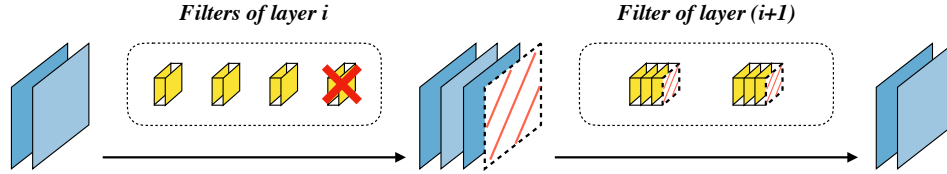

*Filters of layer i*        *Filter of layer (i+1)*

Figure 1: An illustration of filter pruning. The $i$-th layer has 4 filters (*i.e.* channels). If we remove one of the filters, the corresponding feature map will disappear, and the input of the filters in the $(i + 1)$-th layer changes from 4 channels to 3 channels.

memory to store the intermediate output, which is a heavy burden for the low-end devices. Therefore, network compression and acceleration methods have aroused great interest in research.

Recent studies on model compression and acceleration can be divided into four categories: 1) Quantization [34, 55, 54]. 2) Fast convolution [2, 41]. 3) Low rank approximation [7, 8, 50]. 4) Filter pruning [1, 30, 15, 25, 28, 33, 56, 52]. Among these methods, filter pruning (a.k.a. channel pruning) has received widespread attention due to its notable advantages. First, filter pruning is a universal technique that can be applied to various types of CNN models. Second, filter pruning does not change the design philosophy of the model, which makes it easy to combine with other compression and acceleration techniques. Furthermore, no specialized hardware or software is needed for the pruned network to earn acceleration.

Neural pruning was first introduced by Optimal Brain Damage (OBD) [23, 10], in which LeCun *et al.* found that some neurons could be deleted without noticeable loss in accuracy. For CNNs, we prune the network at the filter level, so we call this technique Filter Pruning (Figure 1). The studies on filter pruning can be separated into two classes: 1) Layer-by-layer pruning [30, 15, 56]. 2) Global pruning [25, 28, 33, 52]. The layer-by-layer pruning approaches remove the filters in a particular layer at a time until certain conditions are met, and then minimize the feature reconstruction error of the next layer. But pruning filters layer by layer is time-consuming especially for the deep networks. Besides, a pre-defined pruning ratio required to be set for each layer, which eliminates the ability of the filter pruning algorithm in neural architecture search, we will discuss it in the section 4.3. On the other hand, the global pruning method removes unimportant filters, no matter which layer they are. The advantage of global filter pruning is that we do not need to set the pruning ratio for each layer. Given an overall pruning objective, the algorithm will reveal the optimal network structure it finds. The key to global pruning methods is to solve the global filter importance ranking (GFIR) problem.

In this work, we propose a novel global filter pruning method, which includes two components: The first is the Gate Decorator algorithm to solve the GFIR problem. The second is the Tick-Tock pruning framework to boost pruning accuracy. Specially, we show how to apply the Gate Decorator to the Batch Normalization [19], and we call the modified module Gate Batch Normalization (GBN). It should be noted that the modules transformed by Gate Decorator are designed to serve the temporary purpose of pruning. Given a pre-trained model, we convert the BN modules to GBN before pruning. When the pruning ends, we turn GBN back to vanilla BN. In this way, no special operations or structures are introduced. The extensive experiments demonstrate the effectiveness of our approach. We achieve the state-of-the-art pruning ratio on ResNet-56 [11] by reducing 70% FLOPs without noticeable loss in accuracy. On ImageNet [4], we reduce 40% FLOPs of the ResNet-50 [11] while increase the top-1 accuracy by 0.31%. Our contributions can be summarized as follows:

(a) We propose a global filter pruning pipeline, which is composed of two parts: One is the Gate Decorator algorithm designed to solve the GFIR problem, and the other is the Tick-Tock pruning framework to boost pruning accuracy. Besides, we propose the Group Pruning technique to solve the Constraint Pruning problem encountered when pruning the network with shortcuts like ResNet [11].

(b) Experimental results show that our approach outperforms state-of-the-art methods. We also extensively study the properties of the GBN algorithm and the Tick-Tock framework. Furthermore, we demonstrate that the global filter pruning method can be viewed as a task-driven network architecture search algorithm.

## 2 Related work

**Filter Pruning** Filter pruning is a promising solution to accelerate CNNs. Numerous inspiring works prune the filters by evaluating their importance. Heuristic metrics are proposed, such as the magnitude of convolution kernels [25], the average percentage of zero activations (APoZ) [17]. Luo *et al.* [30] and He *et al.* [15] use Lasso regression to select the filters that minimize the next layer's feature reconstruction error. Yu *et al.* [52], on the other hand, optimizes the reconstruction error of the final response layer and propagates the importance score for each filter. Molchanov *et al.* [33] applies Taylor expansion to evaluate the effect of filters on the final loss function. Another category of works trains the network under certain restrictions, which zero out some filters or find redundancy in them. Zhuang *et al.* [56] get good results by applies additional discrimination-aware losses to fine-tune the pre-trained model and keep the filters that contribute to the discriminative power. However, the discrimination-aware losses are designed for classification tasks, which limits its scope of application. Liu *et al.* [28] and Ye *et al.* [49] apply scaling factors to each filter and add the sparse constraint to the loss in the training or fine-tuning stage. Ding *et al.* [6] proposes a new optimization method that forces several filters to reach the same value after training, and then safely removes the redundant filters. These methods need to train the model from scratch, which can be time-consuming for large data sets.

**Other Methods** Quantization methods compress the network by reducing the number of different parameter values. [34, 3] quantize the 32-bit floating point parameter into binary or ternary. But these aggressive quantitative strategies usually comes with accuracy loss. [55, 54] show that when using a moderate quantification strategy, the quantification network can even outperform the full precision network. Recently, new designs of convolution are proposed. Chen *et al.* [2] designed a plug-and-play convolutional unit named OctConv, which factorizes the mixed feature maps by their frequencies. Experimental results demonstrate that OctConv can improve the accuracy of the model while reducing the calculation. Low-rank decomposition methods [7, 8, 5] approximate network weights with multiple lower rank matrices. Another popular research direction for accelerating network is to explore the design of network architecture. Many computation-efficient architectures [16, 36, 53, 32] are proposed for mobile devices. These networks are designed by human experts. To combine the advantages of computers, automatic neural structure search (NAS) has recently received widespread attention. Many studies has been proposed, including reinforcement-learning-based [57], gradient-based [47, 27], evolution-based [35] methods. It should be noted that the Gate Decorator algorithm we proposed is orthogonal to the methods described in this subsection. That is, Gate Decorator can be combined with these methods to achieve higher compression and acceleration rates.

## 3 Method

In this section, we first introduce the Gate Decorator (GD) to solve the GFIR problem. And show how to apply GD to the Batch Normalization [19]. Then we propose an iterative pruning framework called Tick-Tock for better pruning accuracy. Finally, we introduce the Group Pruning technique to solve the Constraint Pruning problem encountered when pruning the network with shortcuts.

### 3.1 Problem Definition and Gate Decorator

Formally, let $\mathcal{L}(X, Y; \theta)$ denotes the loss function used to train the model, where $X$ is the input data, $Y$ is the corresponding label, and $\theta$ is the parameters of the model. We use $\mathcal{K}$ to represent the set of all filters of the network. Filter pruning is to choose a subset of filters $k \subset \mathcal{K}$ and remove their parameters $\theta_k^-$ from the network. We note the left parameters as $\theta_k^+$, therefore we have $\theta_k^+ \cup \theta_k^- = \theta$. To minimize the loss increase, we need to carefully choose the $k^*$ by solving the following optimization problem:

$$k^* = \arg\min_k \left| \mathcal{L}(X, Y; \theta) - \mathcal{L}(X, Y; \theta_k^+) \right| \quad s.t. \ \|k\|_0 > 0 \qquad (1)$$

where $\|k\|_0$ is the number of elements of $k$. A simple way to solve this problem is to try out all possibility of $k$ and choose the best one that has the least effect on the loss. But it needs to calculate the $\Delta\mathcal{L} = \left| \mathcal{L}(X, Y; \theta) - \mathcal{L}(X, Y; \theta_k^+) \right|$ for $\|\mathcal{K}\|_0$ times to do just one iteration of pruning, which is infeasible for deep models that has tens of thousands of filters. To solve this problem, we propose the Gate Decorator to evaluate the importance of filters efficiently.

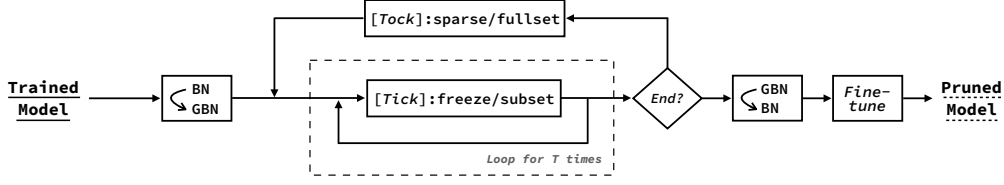

Figure 2: An illustration of the Tick-Tock pruning framework. The Tick phase is executed on a subset of the training data, and the convolution kernels are set to non-updatable. The Tock uses the full training data and adds the sparse constraint of $\phi$ to the loss function.

Assuming that feature map $z$ is the output of the filter $k$, we multiply $z$ by a trainable scaling factor $\phi \in \mathbb{R}$ and use $\hat{z} = \phi z$ for further calculations. When the gate $\phi$ is zero, it is equivalent to pruning the filter $k$. By using Taylor expansion, we can approximately evaluate the $\Delta \mathcal{L}$ of the pruning. Firstly, for notation convenience, we rewrite the $\Delta \mathcal{L}$ in Eq. (2), in which $\Omega$ includes $X$, $Y$ and all of the model parameters except $\phi$. Hence $\mathcal{L}_\Omega(\phi)$ is a unary function *w.r.t* $\phi$.

$$\Delta \mathcal{L}_\Omega(\phi) = |\mathcal{L}_\Omega(\phi) - \mathcal{L}_\Omega(0)| \tag{2}$$

Then we use the Taylor series to expand $\mathcal{L}_\Omega(0)$ in Eq. (3-4),

$$\mathcal{L}_\Omega(0) = \sum_{p=0}^{P} \frac{\mathcal{L}_\Omega^{(p)}(\phi)}{p!} (0 - \phi)^p + R_P(\phi) \tag{3}$$

$$= \mathcal{L}_\Omega(\phi) - \phi \nabla_\phi \mathcal{L}_\Omega + R_1(\phi) \tag{4}$$

Combine Eq. (2) and Eq. (4), we get

$$\Delta \mathcal{L}_\Omega(\phi) = |\phi \nabla_\phi \mathcal{L}_\Omega - R_1(\phi)| \approx |\phi \nabla_\phi \mathcal{L}_\Omega| = \left| \frac{\delta \mathcal{L}}{\delta \phi} \phi \right| \tag{5}$$

$R_1$ is the Lagrange remainder, and we neglect this term because it requires a massive amount of calculation. Now, we are able to solve the GFIR problem base on Eq. (5), which can be easily computed during the process of back-propagation. For each filter $k_i \in \mathcal{K}$, we use $\Theta(\phi_i)$ calculated by Eq. (6) as its importance score, where $\mathcal{D}$ is the training set.

$$\Theta(\phi_i) = \sum_{(X,Y) \in \mathcal{D}} \left| \frac{\delta \mathcal{L}(X, Y; \theta)}{\delta \phi_i} \phi_i \right| \tag{6}$$

Specially, we apply the Gate Decorator to the Batch Normalization [19] and use it for our experiments. We call the modified module Gated Batch Normalization (GBN). We chose the BN module for two reasons: 1) The BN layer follows the convolution layer in most cases. Hence we can easily find the correspondence between the filters and the feature maps of the BN layer. 2) We can take advantage of the scaling factor $\gamma$ in BN to provide the ranking clue for $\phi$ (see Appendix A for details). GBN is defined in Eq. (7), in which $\vec{\phi}$ is a vector of $\phi$ and $c$ is the channel size of $z_{in}$. Moreover, for the networks that do not use BN, we can also directly apply the Gate Decorator to the convolution. The definition of Gated Convolution can be seen at Appendix B.

$$\hat{z} = \frac{z_{in} - \mu_\mathcal{B}}{\sqrt{\sigma_\mathcal{B}^2 + \epsilon}}; \quad z_{out} = \vec{\phi}(\gamma \hat{z} + \beta), \ \vec{\phi} \in \mathbb{R}^c \tag{7}$$

## 3.2 Tick-Tock Pruning Framework

In this section, we introduce an iterative pruning framework to improve pruning accuracy, which called Tick-Tock (Figure 2). The Tick step is designed to achieve following goals: 1) Speed up the pruning process. 2) Calculate the importance score $\Theta$ of each filter. 3) Fix the internal covariate shift problem [19] caused by previous pruning. In the Tick phase, we train the model on a small subset of the training data for one epoch, in which we only allow the gate $\phi$ and the final linear layer to be updatable to avoid overfitting on the small dataset. $\Theta$ is calculated during the backward propagation according to Eq. (6). After training, we sort all the filters by their importance score $\Theta$ and remove a portion of the least important filters.

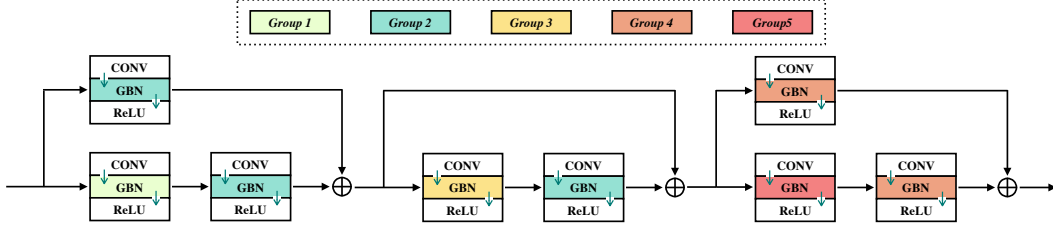

Figure 3: An illustration of Group Pruning. GBNs with the same color belong to the same group.

The Tick phase could be repeated $T$ times until the Tock phase comes in. The Tock phase is designed to fine-tune the network to reduce the accumulation of errors caused by removing filters. Besides, a sparse constraint on $\phi$ is added to the loss function during training, which helps to reveal the unimportant filters and calculate $\Theta$ more accurately. The loss used in Tock is shown in Eq. (8).

$$\mathcal{L}_{tock} = \mathcal{L} + \lambda \sum_{\phi \in \Phi} |\phi| \tag{8}$$

Finally, we fine-tune the pruned network to get better performance. There are two differences between the Tock step and the Fine-tune step: 1) Fine-tune usually trains more epochs than Tock. 2) Fine-tune does not add the sparse constraint to the loss function.

## 3.3 Group Pruning for the Constrained Pruning Problem

ResNet [11] and its variants [18, 46, 43] contain *shortcut connections*, which applies element-wise addition on the feature maps that produced by two residual blocks. If we prune the filters of each layer independently, it may result in the misalignment of feature maps in the shortcut connection.

Several solutions are proposed. [25, 30] bypass these troublesome layers and only prune the internal layers of the residual blocks. [28, 15] insert an additional sampler before the first convolution layer in each residual block and leave the last convolution layer unpruned. However, avoiding the troublesome layers limits the pruning ratio. Moreover, the sampler solution adds new structures to the network, which will introduce additional computational latency.

To solve the misalignment problem, we propose the Group Pruning: we assign the GBNs connected by the *pure shortcut connections* to the same group. The pure shortcut connection is a shortcut with no convolution layer on the side branch, as shown in Figure 3. A group can be viewed as a Virtual GBN that all its members share the same pruning pattern. And the importance score of the filters in the group is the sum of its members, as shown in Eq. (9). $g$ is one of the GBN members in the group $G$, and the ranking of $j$-th filter of all members in $G$ is determined by $\Theta(\phi_j^G)$.

$$\Theta(\phi_j^G) = \sum_{g \in G} \Theta(\phi_j^g) \tag{9}$$

## 3.4 Compare to the Similar Work.

PCNN [33] also uses the Taylor expansion to solve the GFIR problem. The proposed Gate Decorator differs from PCNN in three aspects: 1) Since no scaling factors are introduced, PCNN evaluates the filter's importance score by summing the first degree Taylor polynomials of each element in its feature map, which will accumulate the estimation error. 2) Moreover, PCNN could not take advantage of the sparse constraint due to the lack of scaling factor. However, according to our experiments, sparse constraint plays an important role in boosting the pruning accuracy. 3) A score normalization across layers is essential for PCNN, but not for Gate Decorator. This is because PCNN uses the accumulation method to calculate the importance score, which will lead to the scale of scores varies with the size of feature maps across layers. We abandon the score normalization since our scores are globally comparable, and normalization will introduce new estimation errors.

# 4 Experiments

In this section, we first introduce the datasets and general implementation details used in our experiments. We then confirmed the effectiveness of the proposed method by comparing it with several state-of-the-art approaches. Finally, we explore in detail the role of each component.

## 4.1 Implementation Details

**Datasets.** Various datasets are used in our experiments, including CIFAR-10 [20], CIFAR-100 [20], CUB-200 [45], ImageNet ILSVRC-12 [4] and PASCAL VOC 2011 [31]. The CIFAR-10 [20] dataset contains 50K training images and 10K test images for 10 classes. The CIFAR-100 [20] dataset is just like the CIFAR-10, except it has 100 classes containing 600 images each. The CUB-200 [45] dataset consists of nearly 6,000 training images and 5,700 test images, covering 200 birds species. The ImageNet ILSVRC-12 [4] contains 1.28 million training images and 50K test images for 1000 classes. The PASCAL VOC 2011 [31] segmentation dataset and its extended dataset SBD [9] are used, which provides 8,498 training images and 2,857 test images in 20 categories.

**Baseline training.** Three types of popular network architectures are adopted: VGGNet [39], ResNet [11] and FCN [38]. Since the VGGNet is originally designed for the ImageNet classification tasks, for the CIFAR and CUB-200 tasks, we use the full convolution version of the VGGNet taken from [22] which we note as VGG-M. All networks are trained using SGD, with weight decay and momentum set to $10^{-4}$ and 0.9, respectively. We train our CIFAR and ImageNet baseline models by following the setup in [11]. For CIFAR datasets, the model was trained for 160 epochs with a batch size of 128. The initial learning rate is set to 0.1 and divide it by 10 at the epoch 80 and 120. Besides, the simple data augmentation addressed in [11] is also adopted: random crop and random horizontal flip the training images. For ImageNet, we trained the baseline model for 90 epochs with a batch size of 256. The initial learning rate is set to 0.1 and divide it by 10 every 30 epochs. We follow the widely used data augmentation in [21]: images are resized to $256\times256$, then randomly crop a 224x224 area from the original image or its horizontal reflection for training. The testing is on the center crop of $224\times224$ pixels. For the semantic segmentation task, We train an FCN-32s [38] network taken from [44] for 11 epoch.

**Tick-Tock settings.** Since ResNet is more compact than VGG, we prune 0.2% filters of ResNet and 1% filters of VGG (including FCN) in each Tick stage. The Tick stage can be performed based on a subset of the training data to speed up the pruning. For the ImageNet task, we randomly draw 100 images per class to form the subset. In the case of CIFAR and CUB-200, we use all training data in Tick due to the small scale of the dataset. In all of our experiments, $T$ is set to 10, which means that one Tock operation is performed after every 10 Tick operations. And we train the network with sparse constraint for 10 epochs in the Tock phase. If not stated otherwise, we use the following learning rate adjustment strategy. The learning rate used in Tick is set to $10^{-3}$. For the Tock phase, we use the 1-cycle [42] strategy to linearly increases the learning rate from $10^{-3}$ to $10^{-2}$ in the first half of the iteration, and then linearly decrease from $10^{-2}$ to $10^{-3}$. For the Fine-tune phase, we use the same learning rate strategy as the Tock phase to train the network for 40 epochs.

## 4.2 Overall Comparisons

**ResNet-56 on the CIFAR-10.** Table 1 shows the pruning results of ResNet-56 on CIFAR-10. We compare GBN with various pruning algorithms, and we can see that GBN has achieved the state-of-the-art pruning ratio without noticeable loss in accuracy. Our pruned ResNet-56 with 60% FLOPs

| Metric | Li et al. [25] | NISP [52] | DCP-A [56] | CP [15] | AMC [14] | C-SGD [6] | GBN-40 | GBN-30 |
|---|---|---|---|---|---|---|---|---|
| FLOPs ↓% | 27.6 | 43.6 | 47.1 | 50.0 | 50.0 | 60.8 | **60.1** | **70.3** |
| Params ↓% | 13.7 | 42.6 | 70.3 | - | - | - | **53.5** | **66.7** |
| Accuracy ↓% | -0.02 | 0.03 | -0.01 | 1.00 | 0.90 | -0.23 | **-0.33** | **0.03** |

Table 1: The pruning results of ResNet-56 [11] on CIFAR-10 [20]. The baseline accuracy is 93.1%.

Table 2: The pruning results of ResNet-50 [11] on the ImageNet [4] dataset. "P.Top-1" and "P.Top-5" denotes the top-1 and top-5 single center crop accuracy of the pruned model on the validation set. "[Top-1] ↓" and "[Top-5] ↓" denotes the decrease in accuracy of the pruned model compared to its unpruned baseline. "Global" identifies whether the method is a global filter pruning algorithm.

| Method | Global | P. Top-1 | [Top-1] ↓ | P. Top-5 | [Top-5] ↓ | FLOPs ↓% | Param ↓% |
|---|---|---|---|---|---|---|---|
| ThiNet-70 [30] | ✗ | 72.04 | 0.84 | 90.67 | 0.47 | 36.75 | 33.72 |
| SFP [12] | ✗ | 74.61 | 1.54 | 92.06 | 0.81 | 41.80 | - |
| **GBN-60** | ✓ | **76.19** | **-0.31** | **92.83** | **-0.16** | **40.54** | **31.83** |
| NISP [52] | ✓ | - | 0.89 | - | - | 44.01 | 43.82 |
| FPGM [13] | ✗ | 74.83 | 1.32 | 92.32 | 0.55 | 53.50 | - |
| ThiNet-50 [30] | ✗ | 71.01 | 1.87 | 90.02 | 1.12 | 55.76 | 51.56 |
| DCP [56] | ✗ | 74.95 | 1.06 | 92.32 | 0.61 | 55.76 | 51.45 |
| GDP [26] | ✓ | 71.89 | 3.24 | 90.71 | 1.59 | 51.30 | - |
| **GBN-50** | ✓ | **75.18** | **0.67** | **92.41** | **0.26** | **55.06** | **53.40** |

reduction outperforms the baseline by 0.33% in the test accuracy. When reducing the FLOPs by 70%, the test accuracy is only 0.03% lower than the baseline model.

**ResNet-50 on the ILSVRC-12.** To validate the effectiveness of the proposed method on large-scale datasets, we further pruning the widely used ResNet-50 [11] on the ILSVRC-12 [4] dataset. The results are shown in Table 2. We fine-tune the pruned network for 60 epochs with a batch size of 256. The learning rate is initially set to 0.01 and divided by 10 at epoch 36, 48 and 54. We also test the acceleration of the pruned networks in wall clock time. The inference speed of the baseline ResNet-50 model on a single Titan X Pascal is 864 images per second, using batch size of 64 and image resolution of 224×224. By reducing 40% FLOPs, GBN-60 can process 1127 images per second (30%↑). And GBN-50 achieves 1237 images per second (43%↑).

**FCN on the PASCAL VOC 2011.** Since the Gate Decorator does not make any assumptions about the loss function and no additional operation or structures are introduced into the pruned model, the proposed method can be easily applied to various computer vision tasks. We tested the effect of the proposed method on the semantic segmentation task by pruning an FCN-32s [38] network on the extended PASCAL VOC 2011 dataset [31, 9] (see Appendix C for details). Compared to the baseline, The pruned network reduces the FLOPs by 27% and the parameters amount by 73% while maintain the mIoU (62.84%→62.86%).

## 4.3 More Explorations

**Comparisons on the GFIR problem.** To verify the effectiveness of Gate Decorator in solving the global filter importance ranking problem, we compare it with the other two global filter pruning methods, Slim [28] and PCNN [33]. Concerning the baseline model, we employ the VGG-16-M [22] model that pre-trained on the ImageNet [4] and train it on the CUB-200 [45] dataset for 90 epochs with batch size 64. The initial learning rate is set to $3 \times 10^{-3}$ and divide it by 3 every 30 epochs. All pruning tests are based on the same baseline model. The sparse constraints on the scale factors are adopted except for PCNN in the Tock phase, and the $\lambda$ for the sparse constraint is set to $10^{-3}$.

From the results shown in Figure 4, we have the following observations: 1) The accuracy of the model pruned by Slim [28] changes dramatically. It is because the Slim method ranks filters by the magnitude of its factors, which is insufficient according to our analysis. Besides, Slim can also benefit from the sparse constraint in the Tock phase. 2) The PCNN [33] and the proposed GBN produce a smoother curve due to the gradient is taken into account. GBN outperforms PCNN by a large margin due to the differences discussed in Section 3.4.

**Global filters pruning as a NAS method.** The global filter pruning method automatically determines the channel size of each layer of the CNN model, which can cooperate with the popular NAS methods [47, 27, 57, 35] to obtain more efficient network architecture for a specific task. Figure 5 shows two compressed networks with the same amount of computation. The baseline model is a VGG-16-M network trained on the CIFAR-100 [20] with an test accuracy of 73.19%. The "shrunk"

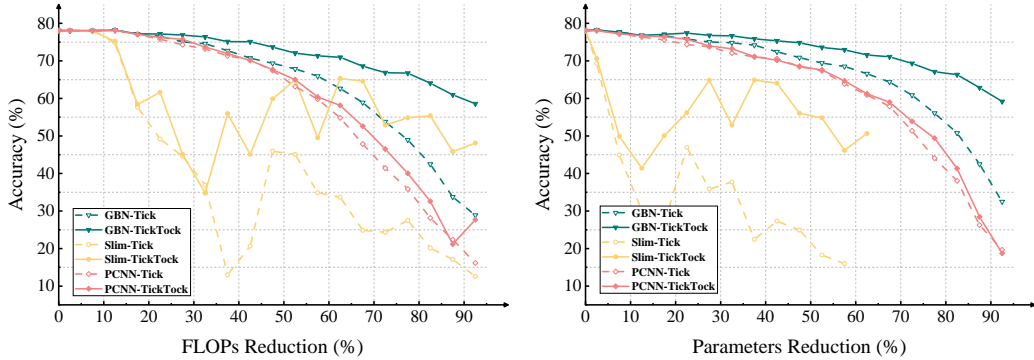

Figure 4: The pruning results of VGG-16-M [22] on the CUB-200 [45] dataset. The reported results are the model test accuracy before the Fine-tune phase. Slim [28] and PCNN [33] are compared.

network halve the channel size of all convolution layers, so its FLOPs become 1/4 of the baseline. We train the "shrunk" network for 320 epochs from scratch, and the test accuracy dropped by 1.98% compared to the baseline. The "pruned" network is the result of pruning the baseline model using Tick-Tock framework, which only drops the accuracy by 1.30%. If we reinitialize the "pruned" network and train it from scratch, the accuracy can reach 71.02%. More importantly, the number of parameters of the "pruned" network is only 1/3 of the "shrunk" one. Comparing their structure, we find that the redundancy in the deep layers is unnecessary, while the middle layers seem to be more important, which is different from our knowledge. Therefore, this experiment demonstrates our pruning method can be viewed as a task-driven network architecture search algorithm, which is also consistent with the conclusion presented in [29].

**Effectiveness of the Tick-Tock framework.** To figure out the impact of the Tick-Tock framework, we integrate GBN with three different pruning schemas. Test accuracy of the pruned models are shown in Table 4.3. In the One-Shot mode, we only calculate the global filter ranking once and prune the model to certain FLOPs without reconstruction. As to the Tick-Only mode, Tick is repeated until the FLOPs of the network falls below a certain threshold. In each Tick step, GBN recalculates the global filter ranking and removing 1% of the filters. We apply the full pruning pipeline in the Tick-Tock mode. Also, due to there are additional training during the Tick-Tock pipeline, we double the epochs to 320 when training the "scratch" model for a fair comparison.

At the same FLOPs threshold, the models pruned by Tick-Tock and Tick-Only are significantly better than One-Shot in both "fine-tune" and "scratch" results, which shows iterative pruning is more accurate. Comparing Tick-Only with Tick-Tock, on the one hand, the accuracy of the models trained from scratch are comparable, because similar network structures are preserved (see Appendix E for details). On the other hand, The Tock phase enhances the performance of the pruned model, which benefits from the sparse constraint. When reducing 40% FLOPs, the pruned model achieves 74.6% accuracy on the test set, which is 1.4% higher than the unpruned model.

| FLOPs- | GBN with One-Shot | | | GBN with Tick-Only | | | GBN with Tick-Tock | | |
|---|---|---|---|---|---|---|---|---|---|
| Pruned | Param | Finetune | Scratch | Param | Finetune | Scratch | Param | Finetune | Scratch |
| 40% | 79.3% | 71.8 | 73.1 | 68.5% | 73.0 | 73.7 | 69.0% | **74.6** | 73.7 |
| 60% | 92.0% | 62.1 | 68.0 | 86.2% | 71.4 | 72.9 | 85.5% | **73.2** | 73.0 |
| 80% | 97.5% | 57.7 | 59.9 | 95.0% | 68.4 | 69.6 | 94.7% | **71.2** | 69.9 |

Table 3: The test results of VGG-16-M [22] model on the CIFAR-100 [20] dataset under different pruning schemas. The accuracy of unpruned baseline model is 73.2%. "Param" denotes the percentage of parameters that been removed. "Finetune" represents the test accuracy of the pruned model after fine-tuning. "Scratch" shows the test result of the random initialized model, which has the same architecture as the pruned one. When training the "Scratch" model, we doubled the epochs to 320.

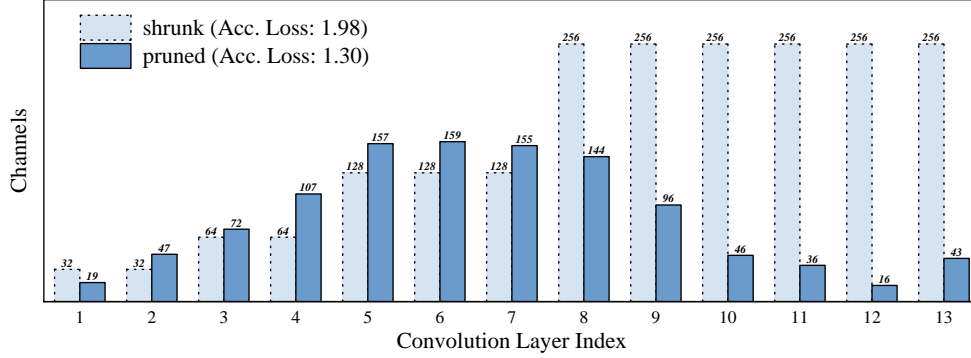

Figure 5: An illustration of two network architectures with the same FLOPs.

# 5 Conclusion

In this work, we propose three components to serve the purpose of global filter pruning: 1) The Gate Decorator algorithm to solve the global filter importance ranking (GFIR) problem. 2) The Tick-Tock framework to boost pruning accuracy. 3) The Group Pruning method to solve the constrained pruning problem. We show that the global filter pruning method can be viewed as a task-driven network architecture search algorithm. Extensive experiments show the proposed method outperforms several state-of-the-art filter pruning methods.

# 6 Acknowledgement

This work was supported by National Key R&D Program of China (Grant no.2017YFB1200700), Peking University Medical Cross Research Seed Fund and National Natural Science Foundation of China (Grant no.61701007).

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
