[Supplementary Material · appendix.pdf]

# Appendix

## A  Details of the transformation between GBN and BN

The definition of GBN is as follows:

$$\hat{z} = \frac{z_{in} - \mu_{\mathcal{B}}}{\sqrt{\sigma_{\mathcal{B}}^2 + \epsilon}}; \quad z_{out} = \vec{\phi}(\gamma\hat{z} + \beta), \ \vec{\phi} \in \mathbb{R}^c$$

According to [26], the magnitude of $\gamma$ provides the information about the filter ranking. We can take advantage of this information through the following transformation. First, we convert the BN to GBN using the following formula:

$$\vec{\phi} := \gamma$$
$$\beta := \frac{\beta}{\gamma}$$
$$\gamma := 1$$

Then we fix $\gamma$ by setting it to non-updatable during the pruning. When the pruning is finished, we convert GBN back to BN by merging $\vec{\phi}$ into $\gamma$ and $\beta$:

$$\gamma := \vec{\phi}$$
$$\beta := \beta\vec{\phi}$$

## B  The Design of Gated Convolution

Let $W \in \mathbb{R}^{c \times k \times k}$ denote a $k \times k$ convolution kernel of a filter, and $X, Y \in \mathbb{R}^{c \times h \times w}$ denote the input and output tensors, respectively. We note the convolution operation as:

$$Y = X \otimes W$$

We use the following formula to convert the filter to the gated version:

$$\phi := \frac{\|W\|_F}{ck^2}$$
$$W := \frac{W}{\phi}$$

And we modify the convolution operation to:

$$Y = \phi(X \otimes W)$$

When the pruning is finished, we merge $\phi$ into $W$:

$$W := \phi W$$

## C  Segmentation Instances

We tested the effect of the proposed method on the semantic segmentation task by pruning an FCN-32s [36] network on the extended PASCAL VOC 2011 dataset [29, 9]. Since there is no BN layer in the FCN-32s network structure, we replace the convolution layer with the Gated Convolution designed in Appendix B. Our code is taken from [42] and follow its default training settings. When

(a) Baseline (mIoU: 62.84)

(b) Pruned (mIoU: 62.88)

Figure 6: The segmentation instances.

training the baseline, the batch size is set to 1, and the learning rate is set to $10^{-10}$. In the TICK phase, the learning rate is set to $10^{-10}$. In the Tock phase, we are using a 1-cycle strategy [40], and the learning rate linearly varies from $10^{-11}$ to $10^{-10}$. In the Fine-tune stage, we still used $10^{-10}$ learning rates to train the network for 10 epochs. Figure 6 shows the segmentation results from (a) the baseline network and (b) the pruned network. The pruned network reduces the FLOPs by 27% and the parameters amount by 73% while maintain the mIoU (62.84%→62.86%)

## D   Structure of Pruned ResNet-50 (GBN-60).

Figure 7: The channel pruning percentage of each convolutional layer (excluding the shortcut) of the GBN-60 network in Table 2. The index "a-b" indicates the residual block in which the convolutional layer is located. For example, "3-6" indicates the convolution layer is located in the conv3_6 [11].

## E   Structures of the Pruned Networks in Table 3

Figure 8 shows the structures of the pruned network in Table 3 at 80% FLOPs reduced. The minimum number of channels is set to 9. We can get the following observations: On the one hand, the One-Shot mode cannot accurately perform network pruning. On the other hand, the network structures obtained by the Tick-Only and Tick-Tock modes is similar.

| One-Shot | Tick-Only | Tick-Tock |
|---|---|---|

```
DataParallel(                        DataParallel(                        DataParallel(
  (module): VGG(                       (module): VGG(                       (module): VGG(
    (features): Sequential(              (features): Sequential(              (features): Sequential(
      (0): Conv2d(3, 25)                   (0): Conv2d(3, 22)                   (0): Conv2d(3, 18)
      (1): BatchNorm2d(25)                 (1): BatchNorm2d(22)                 (1): BatchNorm2d(18)
      (2): ReLU()                          (2): ReLU()                          (2): ReLU()
      (3): Conv2d(25, 61)                  (3): Conv2d(22, 48)                  (3): Conv2d(18, 46)
      (4): BatchNorm2d(61)                 (4): BatchNorm2d(48)                 (4): BatchNorm2d(46)
      (5): ReLU()                          (5): ReLU()                          (5): ReLU()
      (6): MaxPool2d()                     (6): MaxPool2d()                     (6): MaxPool2d()
      (7): Conv2d(61, 76)                  (7): Conv2d(48, 69)                  (7): Conv2d(46, 66)
      (8): BatchNorm2d(76)                 (8): BatchNorm2d(69)                 (8): BatchNorm2d(66)
      (9): ReLU()                          (9): ReLU()                          (9): ReLU()
      (10): Conv2d(76, 126)                (10): Conv2d(69, 97)                 (10): Conv2d(66, 99)
      (11): BatchNorm2d(126)               (11): BatchNorm2d(97)                (11): BatchNorm2d(99)
      (12): ReLU()                         (12): ReLU()                         (12): ReLU()
      (13): MaxPool2d()                    (13): MaxPool2d()                    (13): MaxPool2d()
      (14): Conv2d(126, 83)                (14): Conv2d(97, 130)                (14): Conv2d(99, 133)
      (15): BatchNorm2d(83)                (15): BatchNorm2d(130)               (15): BatchNorm2d(133)
      (16): ReLU()                         (16): ReLU()                         (16): ReLU()
      (17): Conv2d(83, 83)                 (17): Conv2d(130, 124)               (17): Conv2d(133, 132)
      (18): BatchNorm2d(83)                (18): BatchNorm2d(124)               (18): BatchNorm2d(132)
      (19): ReLU()                         (19): ReLU()                         (19): ReLU()
      (20): Conv2d(83, 77)                 (20): Conv2d(124, 120)               (20): Conv2d(132, 127)
      (21): BatchNorm2d(77)                (21): BatchNorm2d(120)               (21): BatchNorm2d(127)
      (22): ReLU()                         (22): ReLU()                         (22): ReLU()
      (23): MaxPool2d()                    (23): MaxPool2d()                    (23): MaxPool2d()
      (24): Conv2d(77, 9)                  (24): Conv2d(120, 103)               (24): Conv2d(127, 117)
      (25): BatchNorm2d(9)                 (25): BatchNorm2d(103)               (25): BatchNorm2d(117)
      (26): ReLU()                         (26): ReLU()                         (26): ReLU()
      (27): Conv2d(9, 9)                   (27): Conv2d(103, 67)                (27): Conv2d(117, 76)
      (28): BatchNorm2d(9)                 (28): BatchNorm2d(67)                (28): BatchNorm2d(76)
      (29): ReLU()                         (29): ReLU()                         (29): ReLU()
      (30): Conv2d(9, 9)                   (30): Conv2d(67, 36)                 (30): Conv2d(76, 38)
      (31): BatchNorm2d(9)                 (31): BatchNorm2d(36)                (31): BatchNorm2d(38)
      (32): ReLU()                         (32): ReLU()                         (32): ReLU()
      (33): MaxPool2d()                    (33): MaxPool2d()                    (33): MaxPool2d()
      (34): Conv2d(9, 9)                   (34): Conv2d(36, 24)                 (34): Conv2d(38, 32)
      (35): BatchNorm2d(9)                 (35): BatchNorm2d(24)                (35): BatchNorm2d(32)
      (36): ReLU()                         (36): ReLU()                         (36): ReLU()
      (37): Conv2d(9, 9)                   (37): Conv2d(24, 30)                 (37): Conv2d(32, 11)
      (38): BatchNorm2d(9)                 (38): BatchNorm2d(30)                (38): BatchNorm2d(11)
      (39): ReLU()                         (39): ReLU()                         (39): ReLU()
      (40): Conv2d(9, 9)                   (40): Conv2d(30, 96)                 (40): Conv2d(11, 41)
      (41): BatchNorm2d(9)                 (41): BatchNorm2d(96)                (41): BatchNorm2d(41)
      (42): ReLU()                         (42): ReLU()                         (42): ReLU()
      (43): MaxPool2d()                    (43): MaxPool2d()                    (43): MaxPool2d()
      (44): AvgPool2d()                    (44): AvgPool2d()                    (44): AvgPool2d()
    )                                    )                                    )
    (classifier): Linear(9, 100)         (classifier): Linear(96, 100)        (classifier): Linear(41, 100)
  )                                    )                                    )
)                                    )                                    )
```

Figure 8: Structures of the pruned networks in Table 3.