[Reviews · NeurIPS 2019]

Reviewer 1



1. Before rebuttal: Technical details are clearly written, easily understood, and reproducible. The experiments can validate the authors' argued contribution. My only concern is that the core novelty of the paper should be formally refined, since Taylor expansion has been used before in [31], Tick-tock is not novel enough compared to previous fine-tuning step, and group pruning is an engineering technique for using the proposed GBN. 2. After rebuttal: Reviewers #2 and #3 also have concerns on novelty. I am satisfied with the authors' feedback on the core novelty, so I increase my score to 7.

Reviewer 2



Originality: Minor. Although the overall Gate Decorator method is effective, most of the individual components have been proposed in previous work. For example, pruning filter based on BN is proposed in [26]; The Taylor expansion formula is proposed in [31]; The iterative pruning setting has been proposed in [8]; The sparse constraint on scaling factor has been proposed in [26], etc. Quality: The technical claim is sound. Extensive experiments make the paper convincing. Clarity: The paper is generally well written except for a few typos. It is also well organized. Significance: Minor or medium. The results are not very surprising. Other researchers may borrow the idea of GBN and Tick-Tock procedure to improve the performance. However, it did not compare with the previous state-of-the-art pruning methods based on AutoML, making the advantage less convincing.

Reviewer 3



- This paper is technically sound and presented clear in general. - The idea of global pruning based on BN is not new. [26], [47], as well as the ''Data-Driven Sparse Structure Selection for Deep Neural Networks'' were leveraging a similar idea. Especially in [47], a similar two-step framework was proposed, which does not have to train from scratch (Inaccurate description in Page 3 row 83-84). It would be great if authors could describe the new insights of using Taylor expansion to estimate the change of loss along with importance of a filter. - Fairly amount of experimental results were presented to demonstrated the effectiveness of the proposed idea. It would be great if authors could compare with [26], [47] and ''Data-Driven Sparse Structure Selection for Deep Neural Networks'' (Huang and Wang, 2017) due to the similarity of these ideas. Given the results presented on ImageNet using ResNet-50, which is 0.31 accuracy improvement and 40% FLOPs reduction, it's difficult to compare with (Huang and Wang, 2017) with results of error 26.8 and 66% reduction. [**Update**] Authors addressed my concerns on the comparison with (Huang and Wang, 2017), but did not address others.

[Author Response · NeurIPS 2019]

We sincerely thank all reviewers for your contributions in reviewing this paper. Your comments are very helpful to
refine this work. We will primarily respond to your concerns about the experiment comparisons and algorithm novelty.

**Q1:** Compare with AutoML based pruning methods like AMC[56] and MetaPruning[57] (Reviewer #2).
It's a very good suggestion. Comparing with these works would help us to demonstrate the potential of GD algorithm in
the network search tasks. We ran a new experiment on the **MobileNet** during this week. Although we were unable to
make carefully hyper-parameter tuning due to the time constraints, we still got comparable results. **Moreover, in order**
**to emphasize that our method has achieved SOTA results, we add more comparisons with the latest CVPR'19**
**papers.** We summarized the new comparisons in the following table, which will be included in the final version. The
symbol "←" indicates using the same network or dataset as its left. Our work is reproducible and the code has been
included in the submission for review. We will open source all of our code if this work is accepted.

| Work | [58] | [59] | [6] | **Ours** | [56] | **Ours** | [56] | [57] | **Ours** |
|---|---|---|---|---|---|---|---|---|---|
| Publish | CVPR'19 oral | CVPR'19 | CVPR'19 | - | ECCV'18 | - | ECCV'18 | arxiv'19 | - |
| Network | ResNet-50 | ← | ← | ← | ResNet-56 | ← | MobileNet | ← | ← |
| Dataset | ImageNet | ← | ← | ← | CIFAR-10 | ← | ImageNet | ← | ← |
| FLOPs ↓ | 53% | 55% | 55% | **55%** | 50% | **60%** | 50% | 50% | **60%** |
| Top-1 Acc. | 74.83 | 71.80 | 74.54 | **75.18** | 91.90 | **93.41** | 70.5 | 70.4 | 70.2 |

**Q2:** Compare with SSS[60] (Reviewer #3).
Great thanks for providing this paper. It's a good work and we will include our comparision with it in the final version.
But there is a mistake in your comment which we have to correct. **The result of "error rate 26.8% with 66% FLOPs**
**reduction" you mentioned in [60] is not from ResNet-50 but ResNeXt-50.** However, [60] does provide the pruning
results of ResNet-50 (Table 2: ResNet-50 → ResNet-26), so we can directly compare with it without adding extra
experiments. [60] prunes 43% FLOPs of the ResNet-50 with 71.82% Top-1 accuracy remained. **We could reach 55%**
**FLOPs reduction with 75.18% Top-1 accuracy, which is significantly better than [60].**

**Q3:** The concern about novelty (Reviewer #1 and #2).
We will explain in detail the novelty and contributions of our work. The GD algorithm is inspired by the previous
publications, especially [26, 31]. They are all excellent works, but we found some weaknesses in their methods. [31]
was published in late 2016, which first applies the Taylor series to the filter pruning task. However, because of the
problems we discussed in the section 3.4, the results [31] presents is not outstanding. **The way it applies Taylor series**
**lead to 2 flaws**: (1) The accumulation of estimation errors. (2) The importance scores of filters between different
layers cannot be directly compared with each other. The first problem was ignored, but the second problem cannot be
overlooked. To fix the second problem, [31] has to introduce the mechanism called "score normalization". In spite
of this, the solution is still not ideal. **We are aware of these two problems** in [31] and avoid them by introducing
the gate factor and modifying the way to applies Taylor expansion formula. In the Figure.4 we can see that even
without considering the other improvements proposed in our paper, just introducing this simple change is enough
for our algorithm to **outperform [31] by a large margin** (57% vs. 45% in accuracy under 70% FLOPs reduction).
This improvement is simple and effective, but to our knowledge, in the past nearly three years, no similar work has
been proposed. One of the reasons could be **the flaws in [31] are easy to be neglected.** So we argue that despite this
improvement shows in simple formation, it's still an important contribution.

On the other hand, [26] inspired us to take advantage of $\gamma$ in the BN layer. [26] relies on the absolute value of $\gamma$ to score
the filters. **This makes it performs terrible when pruning a network that trained without sparse constrained on**
$\gamma$ (See Figure 4). But this situation is often encountered, especially when using the networks that pre-trained for
other tasks. Different from [26], we don't require training the network from scratch in sparse constraints. In all our
experiments, the baseline networks before pruning were normally trained without sparse constraints on $\gamma$. Our advantage
comes from more accurate score estimate and **the specially designed Tick-Tock pruning framework.** Furthermore,
for those network without BN, **GD could be directly applied to the convolution layers (see Appendix).**

**The Tick-Tock and Group Pruning are our originally designed modules.** The Tick-Tock is very efficient for
iterative pruning algorithm. According to our experiments, we can **save 70% of the computation time** compared to
just using fine-tune to get the same results in the ImageNet task. Furthermore, **Group Pruning increases the pruning**
**ratio** in the case of constraints, and **it can also be used by other global pruning methods, not just GBN**.

────────────────────────────────────────

[56] "AMC: AutoML for Model Compression and Acceleration on Mobile Devices.", ECCV 2018
[57] Liu et al. "MetaPruning: Meta Learning for Automatic Neural Network Channel Pruning."
[58] "Filter Pruning via Geometric Median for Deep Convolutional Neural Networks Acceleration.", CVPR 2019
[59] "Towards Optimal Structured CNN Pruning via Generative Adversarial Learning.", CVPR 2019
[60] "Data-Driven Sparse Structure Selection for Deep Neural Networks.", ECCV 2018


[Meta-Review · NeurIPS 2019]

A technically sound approach to pruning networks with experimental evidence of performance, and a clearly written paper.